# Industry Context as an Essential Tool for the Future of Healthy and Safe Work: Illustrative Examples for Occupational Health Psychology from the Hospitality Industry

**DOI:** 10.3390/ijerph182010720

**Published:** 2021-10-13

**Authors:** Kristin A. Horan, Mindy K. Shoss, Cynthia Mejia, Katherine Ciarlante

**Affiliations:** 1Department of Psychology, University of Central Florida, Orlando, FL 32816, USA; Mindy.Shoss@ucf.edu (M.K.S.); Katherine.Ciarlante@ucf.edu (K.C.); 2Rosen College of Hospitality Management, University of Central Florida, Orlando, FL 32819, USA; Cynthia.Mejia@ucf.edu

**Keywords:** industry, context, occupational safety and health, occupational health psychology

## Abstract

Contextual nuance holds value for occupational health and safety, particularly as workplace challenges and solutions become more complex. However, disciplines that inform occupational safety and health vary in the degree to which they target breadth and depth of understanding. The future of work presents challenges related to work, the workplace, and the workforce, and an appreciation of the context of industry will ready researchers and practitioners with the most informed solutions. Broadly developed solutions for future of work challenges may flounder without an appreciation for the context of industry, as evidenced by two examples provided in this review. As occupational safety and health disciplines answer the call provided by the future of work, this review provides an account for the value of industry context and recommendations for achieving both breadth and depth of scientific inquiry and practical reach.

## 1. Introduction

There is increasing recognition that the changing nature of work warrants adaptations in traditional approaches of occupational safety and health (OSH) research and practice in order to protect and promote workplace health, safety, and well-being amidst a backdrop of a changing future of work [1]. Future of work trends have tended to be discussed as applying broadly across work, industry, and temporal contexts. This is particularly the case in the occupational health psychology (OHP) literature. Historical accounts of the development of occupational health psychology note a focus on broad topics that apply to the majority of jobs, including supervision and workplace policies and procedures [2]. This focus on breadth within the OHP literature differentiates it from the general OSH literature, which more frequently considers the context experienced by a subset of workers, such as workers within a single sector of the national economy [3]. In this paper, we explore applications of industry context to promote both breadth and depth in OSH disciplines as they address future of work challenges. We describe future of work challenges and solutions that require an appreciation of context within an illustrative group, the hospitality industry, and in doing so hope to motivate OHP researchers and practitioners to incorporate a focus on the nuance provided by industry more often.

The current paper aims to offer an increased appreciation for context as a meaningful component of OSH/OHP research and practice that may shape both (a) what the future of work looks like and (b) the available solutions to its concomitant psychosocial workplace effects. From a conceptual standpoint, Johns [4] defines context as “situational or environmental stimuli that impinge upon focal actors and are often located at a different level of analysis from those actors”. In that vein, elements of omnibus context (the reporters’ questions of what, where, when, why) have downstream consequences on work, work environment, work design, and organizations [5,6]. Johns [4] points out that elements of context that may have particular impact are those that serve as background variables, for example occupation or demographic makeup. We suggest that industry membership is also one of these variables. Industries/sectors have different work demands and work environments, different organizational structures, different workforce characteristics, different values and priorities, and different regulatory and labor context and standards. As the World Health Organization points out, there is no ‘one size fits all’ approach to occupational safety and health [7]. As such, it is difficult to imagine that solutions to occupational health and safety issues would work well for all industries, even if industries share some cross-cutting psychosocial issues.

Within this paper, we will describe the benefits of accounting for industry context, examine current trends in the incorporation of industry context in the OHP literature, and, using the hospitality industry as an example, describe challenges or solutions to future of work issues that would be better understood or implemented with industry in mind. In doing so, we suggest that a greater consideration of the context provided by industry membership will be beneficial within the OHP literature in order to better align its approach with other OSH-informing disciplines and to ready OHP researchers and practitioners for the complex challenges and nuanced solutions that the future of work will bring.

## 2. Industry as Under-Addressed Context in OHP Research

An industry refers to a distinct grouping of productive or profit-making enterprises, narrower than a sector and broader than occupational classifications. Examples include the healthcare industry, the construction industry, and the retail industry. Agencies, such as NIOSH, recognize the value of accounting for industry in the surveillance and intervention of occupational safety and health [3], offering resources that correspond to common industry classification systems including the Census Industry and Occupational Classification System, the North American Industry Classification System (NAICS), and the Standard Occupational Classification (SOC). Although it is a widely understood term, there is no universal operationalization of the variable of “industry”. For example, NIOSH organizes resources and agendas by sectors that are aggregates of NAICS industry classifications and sector groups. This variability in classification and grouping schemes is generally not of concern given that useful cross-walking tools allow for comparison, such as the NIOSH Industry and Occupational Classification Coding System (NOICCS), so long as industry data are collected and coded in research and practice.

The industry-specific context can influence OSH research and practice in several ways. First, industry shapes the setting in which work typically occurs, tasks that are typically performed, the physical or psychosocial hazards that employees encounter, and the types of intervention methods that are possible or easily integrated into existing work processes. Some industry-related contextual factors may limit what is possible; for instance, the implementation of a sit-stand workstation intervention would not be possible in industries such as transportation [8]. This emphasizes the important point that an intervention developed for one industry may not match the context of another and without literature informing the context of each industry, we may not know how to adapt an intervention to promote safety or health across contexts.

Second, industry membership can provide clues to shared values, widely held assumptions, or norms among groups of employees. These powerful forces could influence both the experience of a work stressor or the welcomed or unwelcomed interventions to address that stressor. For example, the idiom “the customer is always right” may act as a threshold for perceptions of customer mistreatment in the hospitality industry and constrain policy and practice options for addressing this psychosocial stressor. What could be considered a major stressor in another industry might be interpreted as “just part of the job” for a hospitality employee. In an exploratory study using fMRI technology, no differences in stress were observed in survey data from hospitality employees and employees of other industries. However, fMRI activation revealed patterns of habituation to stress from customer mistreatment [9]. In other words, more frequent or severe instances of customer mistreatment would be needed to warrant a higher stress score than participants from other industries. Here, the nuance of industry membership sheds incredible light on the processes of work stress that would otherwise be ignored. Based on these findings, an industry-informed researcher would know to expect higher thresholds for self-reported stress when collecting survey data on customer-related stressors in the hospitality industry and/or account for potential industry-related differences among survey data in a general working population sample.

In another example, despite the prevalence of mental health-focused programming in their workplaces, first responders have been known to access employee assistance programs or other resources at lower rates. This has been attributed to a shared value of “mental toughness” resulting in stigma for those who seek mental health treatment [10]. Accounting for these shared ideas could illuminate attitudinal or behavioral trends occurring within groups of employees. An intervening OSH professional might anticipate these industry-related shared perceptions and might first implement pre-intervention programming targeting social stigma to increase readiness for a treatment-based intervention in the public safety sector [11]. These examples illustrate how shared industry-related norms can influence both the experience of stressors and the success of efforts to address those stressors.

Finally, membership in an industry is known to intersect with other topics of contemporary relevance. The OHP literature is advancing to include topics of broader societal importance into our study of work, such as poverty, health equity, and overlooked working populations [12]. A consideration of industry would further these aims given that there are marked trends regarding wages and benefits by industry [13] and that demographic and socioeconomic trends within occupational safety and health risk often vary together by industry [14]. Factors that shape likelihood to participate in an OHP intervention, such as individual factors, structural conditions, or sociocultural context, can also vary by industry [14]. In other words, industry reflects a point of intersection of multiple individual and group identities including both occupational and cultural factors with meaningful implications for health and safety equity, occupational risk factors, and employee intentions related to safety and health. A more purposeful inclusion of industry context into our research could help OHP understand shared experiences of contemporary constructs within groups of employees, such as within an industry. As these contemporary topics become more important in the future of work [1], we assert that now is the ideal time to address an absence of industry context to better equip OHP researchers and practitioners to respond to the complexities of the changing nature of work and extend the positive implications of OHP to often overlooked or disadvantaged groups.

Surprisingly, despite the presence of industry context within national classification and surveillance systems, the current state of the OHP literature tends not to incorporate a strategic focus on the contextual variable of industry. Our research typically relies on methodologies that collect information from participants, generally working adults in OHP research [15]. Although a methods section will often contain a description of contextual factors associated with data collection, such as the industry to which participants belonged, industry is rarely studied as a central element of the research. A study sample containing one or a few industries can often be viewed as a narrow or atypical sample of convenience in field-based survey research (e.g., refer to Landers & Behrend [16] for helpful recommendations on considering the potential implications associated with single industry samples, such as possible range restriction).

The absence of industry context can be observed in both conceptual/theoretical work and empirical studies within OHP. We reviewed four conceptual models or frameworks on determinants and outcomes of work safety, health, and well-being [7,17,18,19]. Central constructs in each model largely focused on the individual level (e.g., personal resources as determinants; stress and physical health as outcomes) and organizational level (e.g., work environment and work practices as determinants; reduced absenteeism and turnover as outcomes). Only the World Health Organization Healthy Workplace Framework named factors existing outside of the organization, such as enterprise involvement with the community. This category of variables focuses on the efforts made by an organization to follow regulatory mandates in ways that cause no harm to the broader community or assist the community in addressing social issues [7]. The WHO Healthy Workplace Framework also explicitly states that their conceptual model is not a one-size fits all solution and that the realization of a healthy workplace will look different from industry to industry [7]. Given this important point, there is an opportunity to increase the presence of industry-relevant constructs in theoretical frameworks that identify the determinants and outcomes of employee safety, health, or well-being in OHP.

Given that our field prioritizes the building of theory [20] it stands to reason that an absence of industry context in theory would translate to a large proportion of industry-agnostic empirical studies in the OHP literature. To confirm this, we borrowed from content analysis methods of previous attempts to characterize the features of the OHP literature [15]. Specifically, we reviewed work-related articles published in the *Journal of Occupational Health Psychology* and *Stress & Health* from the previous five years (372 articles) and found that 74.5% of articles made no mention of industry, 10% incorporated industry into methodology (meaning that they considered how industry could influence the effectiveness of study design or measures, potentially adapting methods accordingly), 10% incorporated industry into the discussion section, and 4% focused their study on phenomena within a specific industry. Only two articles (0.5%) incorporated the features of an industry into their research question. For example, one study [21] examined the industry-related context of training in nursing and how it might influence an occupational health intervention. In another example, a study on posttraumatic stress among police officers incorporated contextual variables related to the public safety industry into research questions, such as the industry tenure and number of case reports [22]. Based on this cursory characterization of recent OHP literature, industry context is indeed underrepresented in the current state of the OHP literature.

We argue that it is imperative to account for industry context as OSH literature evolves to address future of work challenges. The below sections demonstrate how industry context is particularly relevant for occupational health, safety, and well-being in the future of work. First, we describe the OSH challenges associated with the changes in the workplace, the workforce, and work itself. We then provide examples, drawing specifically from the hospitality industry, of when a future of work challenge necessitates an industry-informed solution as opposed to a solution developed in a broad manner to apply to the general workforce.

## 3. Future of Work Challenges and Solutions in Context

There is general agreement that we are entering a future of work and workplaces that is fundamentally different from work and workplaces of the past several decades [23]. Although these changes are in many regards connected to the rapid rise in technological capabilities, there are also larger questions about the nature, design, and management of work as well as questions about government policy (e.g., policy regarding artificial intelligence (AI), labor market policies) and work-relevant societal issues such as climate change [1,24,25]. Within the field of occupational health, there is a sense that efforts should be made to both (a) try to predict trends in order to better adapt to them and (b) try to intervene, direct, or utilize trends in order to promote safer and healthier work [24,25,26].

Tamers and colleagues [1] provided a useful roadmap of future of work topics with relevance to occupational safety and health. They organized subtopics in terms of relevance to the future workplace, future work, and future workforce. Workplace issues concern the design of organizational policies and practices (e.g., sick leave), as well as organizational structure. For example, one major concern is the rise of precarious work, driven by increasing job insecurity and the growing use of “arms-length” employment contracts such as temporary, contingent, contract, agency, seasonal, and gig work [27,28,29]. These precarious work experiences are associated with poorer health and health inequalities as well as accidents [30,31,32]. Other workplace issues involve the impact of automation on job displacement, design, and quality [12]. Work issues concern changes in work itself, which captures for example, the greater use of AI and other emerging technologies (e.g., 3D printing, exoskeletons) [24]. Finally, workforce issues related to the future of work capture topics such as productive aging, worker financial security, and worker skills.

Although organized in separate “buckets,” Tamers and colleagues [1] emphasize that these issues are interconnected and impact each other. For example, worker re-skilling has been discussed as a potential solution to occupational polarization and job displacement due to AI. At the same time, AI and other emerging technologies have been discussed as both threats and opportunities for organizational design and workforce resilience, depending on how they are used. On the upside, AI can enhance the capabilities for real-time workplace monitoring and adjustment for health exposures and accident risks, as well as free workers from tasks that are monotonous or boring [25,32]. On the downside, AI may replace work or jobs, threaten occupational identities, result in further polarization of work quality, and create distrust through surveillance and opaque decision-making [33,34].

This idea that the future of workplace, work, and workforce decisions can have good or bad implications for occupational health and safety also necessitates and points to the value of a context-specific approach to the future of work. Indeed, Tamers and colleagues [1] note this by pointing out that contextual factors such as disasters, politics, and globalization form the backdrop of the future of work and are expected to have pervasive impacts on future workplace, work, and workforce issues. The COVID-19 pandemic and its ensuing work-related changes and challenges, as well as recent AI legislation by the European Union, provide salient examples of how the macro-context is acting to shape the future of work [35,36,37].

To address OSH challenges, particularly those related to the complexities of the future of work, will require a balance of breadth and depth in solutions. An example can be found in the intervention-focused research priorities described in the National Occupational Research Agenda (NORA). Although there is a need to study cross-cutting issues such as work design, the various constraints, opportunities, and needs from each industry will shape the available solutions and the best manner to implement those solutions. Workplace health promotion has documented the need to adapt interventions for a variety of contextual variables, including business size [38] or the proportion of demographic variables (i.e., gender) or job-related variables (i.e., blue collar vs. white collar) that have been shows to correlate with intervention participation [39]. Industry membership shapes experiences of these variables, resulting in barriers to or facilitators for intervention participation [14], and therefore interventions will likely benefit from a proactive consideration of industry context.

In as much as industry is relevant to explain current occupational health and safety issues, it is also a critical lens to apply to future of work issues. For instance, it is likely that disruption due to automation will not be felt equally across industries, and emerging technologies are likely to be implemented and received in different ways depending on industry context. Likewise, solutions to work design issues (e.g., burnout and stress prevention) are likely to be very different when one is dealing with an industry with a large proportion of contingent or agency workers versus an industry where most employment tends to be permanent, full-time. Industries also likely have different opportunities and constraints that may shape how they implement emerging solutions to occupational health and safety challenges. For example, while exoskeletons may be more accepted by the public and workers in nursing or construction, they are unlikely to be accepted in certain core roles in hospitality and entertainment, such as guest-facing roles. As research on AI has demonstrated, occupational identity is critical to how workers react and whether they utilize these systems [40,41]. Even similar OSH solutions may need to look different across different industries. For instance, supply delivery robots may need to operate differently in the hospitality sector where new guests and seasonal staff will interact with the robots than in the healthcare sector where the staff is relatively constant, and robots may not need to interact with the public. The expectations for what these robots look like and how they function is also likely to be quite different across industries. In the following sections, we introduce the hospitality context as one example of how an industry context can better help understand, predict, and intervene in the future of occupational health and safety.

## 4. The Hospitality Industry

The U.S. leisure and hospitality supersector is classified under the service-providing industries supersector, according to the North American Industry Classification System (NAICS) [42]. The Leisure and Hospitality supersector consists of (1) arts, entertainment, and recreation; and (2) accommodation and food services sub-sectors. One of the largest workforce sectors in the U.S. pre-COVID-19, the hospitality industry employed 16.9 million workers in February 2020 [42]. In the post-pandemic workforce environment, the hospitality industry has made steady progress back to what would be considered almost “normal”, reaching employment of 15.2 million workers as of July 2021 [42].

The hospitality industry is consumer-centric, and thus, service quality is a point of brand differentiation and competitive advantage. In the hospitality context, services are consumer-facing (visible), and yet also invisible due to the structures and processes unseen which support service interactions. This hospitality industry ecosystem is one with multiple stakeholders at various levels. Often characterized by “front of house” and “back of house” employees, the former are customer-facing, whereas the latter work in human resources, marketing, sales, and other supportive roles. In addition to the distinction between customer-facing and supporting employees, there is also a stratification of those in management. Typically, line-level employees serve customers, while supervisors, managers, and directors in hospitality organizations occupy the managerial strata and support line level workers. Yet, the ultimate stakeholder is the customer, and successful and well-known hospitality organizations, such as the Ritz-Carlton for example, pride themselves on both external (employee–customer) and internal (employee–employee) service quality, illustrated in their brand motto: “We are Ladies and Gentlemen serving Ladies and Gentlemen” [43].

While the accommodation and foodservice sub-sectors are among the top employers in the U.S. historically, they also lead the U.S. workforce in employee turnover. Prior to the pandemic, the U.S. fast food industry reached turnover rates at nearly 130% [44]. Dubbed the “fast-food turnover crisis”, many brands moved to systemize jobs and thus adopted customer self-service kiosks in the front of the house, and cooking robots in the back of the house as a business strategy to reduce the costs associated with high turnover [44]. Line-level restaurant jobs have been criticized as severely underpaid, intensifying high turnover trends in the industry. Exacerbated by COVID-19 furloughs and layoffs, resulting from mandated shutdowns and stay-at-home orders, the U.S. restaurant industry has struggled to recruit workers as a consequence of not paying workers a fair wage historically, and due to poor treatment during the layoff period [45]. Even though the hospitality industry recovered to report 40% of the total job gains in June 2021, restaurants still face a labor shortage [45] despite adding benefits and increasing wages of up to USD 15 per hour [46]. The U.S. hotel industry also faces similar labor shortages as a result of poor wages and lack of benefits, aggravated by COVID-19 concerns around health and safety, customer mistreatment, and excessive work hours due to co-worker absenteeism or labor shortages [47].

Pressures in the hospitality industry to “get back to normal” compound post-pandemic labor shortages. Those workers who have returned are often income insecure and minority workers who have few options to earn an income, and thus disproportionately suffer from post-pandemic socioeconomic and health disparities [48]. Hotel workers, including housekeepers, laundresses, front desk agents, waiters/waitresses and kitchen workers, are subjected to a variety of physical, biological, chemical, and environmental hazards while on the job [49]. Time pressures to serve the customer, long and irregular hours, exposures to chemical and biological toxins, and work-home conflicts, are all hazards encountered by line-level hotel employees [48,50,51]. Workers in the restaurant and foodservice industry experience similar mental and physical strains on their health and safety, contributing to allostatic load [48]. Chronic and acute work stress experienced by hospitality workers, coupled with their special circumstances around income insecurity, are contributing factors to a variety of health issues and diseases, magnified by the effects of the COVID-19 pandemic [52].

Until recently, the hospitality industry has not placed a high priority on employee health and wellness, aside from those wellness initiatives which generate a rapid return on investment [51]. This is due, in part, to resource allocation associated with thin profit margins, in which a misinformed rationale has undermined justification for employee wellness on its own merits [53,54]. Collectively, the U.S. hospitality industry is facing increasing uncertainties about its labor force in a time of heightened awareness around fair wages and forecasted increasing consumer demand [47]. In the short-term, the hospitality industry may continue to offer higher living wages; however, a more holistic approach would be to offer healthcare, childcare, fertility assistance, tuition reimbursement, career training and development, housing, flexible work schedules, etc. Until the hospitality industry can make substantial progress on much-needed structural changes, the workforce will be hesitant to return. If the current trend continues, workers will continue to seek out employment in other service-related industries, such as in digital sales, shipping, healthcare, and senior living [55]. In the meantime, industry leaders have become increasingly reliant on automation, AI, and other technological solutions to meet the post-COVID-19 consumer demand [56]. With a foundational knowledge of the defining characteristics of the hospitality industry, we will now describe challenges or solutions related to the future of work, both outside of and within the context of the hospitality industry.

## 5. General Future of Work Challenges and Solutions in the Hospitality Context

As noted above, the central thesis of this paper is that as researchers and practitioners respond to the need to align research and interventions to the changing nature of work, they should consider the addition of a meaningful focus on contextual variables (such as industry) as a component of this evolution. We offer two illustrations to describe how future of work-related OSH interventions need to consider industry context. We describe two occupational safety and health challenges created or influenced by future of work trends and their solutions according to broadly-based OHP literature. Specifically, we discuss the challenge of technological job displacement (along with the associated solution of reskilling) and the challenge of increased work-family conflict (and the associated solution of work-family programming). We then describe features within our example industry, the hospitality industry, that would significantly hinder the effectiveness of each solution. Finally, we offer suggestions for a tailored solution that incorporates industry context.

### 5.1. Technological Job Displacement

Anecdotal reports have suggested that by 2025, 85 million jobs will be disrupted globally due to what is known as the “next wave of automation”, largely induced by the pandemic [56] and further spurred by decreasing costs in automation technologies [57]. Yet, scholars in this area offer a different point of view, based on targeted “right” types of technological advancements in the workplace, which hold promise for potentially better social and economic outcomes [58]. The latter argues for new types of jobs or re-skilling due to a technological paradigm shift in the workplace [59]. Regardless, the “changing workplace” category of the future of work paradigm carries the likelihood of “significant downstream effects” including the polarization of skills levels and the threat to middle-skill jobs [1]. The occupational safety and health challenges associated with technological job displacement are clear, as long-term economic and career related outcomes have been noted following job displacement [60]. Related to the economic impacts, job displacement could result in the disruption of benefits needed to maintain health, such as medical insurance, possibly explaining the link between job displacement and mortality [61]. A traditional, broadly-based solution to the future of work challenge of technological job displacement is the reskilling of workers whose medium-skilled jobs will disappear with the goal of making them competitive for a different, higher-skill job [62].

The trend of technological job displacement is present within the hospitality industry but associated with a few contextual nuances. For example, the U.S. hotel industry has not only adopted self-serve kiosks for check-ins, smart device integrations for payment and room key, and service robots for small deliveries, but the industry has also taken the additional step of structurally and architecturally designing hotels for an end-to-end technological experience [57].

In an industry highly dependent on the service experience, critics have called into question the capability of a fully implemented technological guest experience. Moreover, concerns have grown regarding worker displacement and the need for reskilling the hospitality workforce to either work with, or manage, service robots and technologies. Advocates for using more and greater technological resources in the hospitality industry include contactless options for payment and service for health and safety purposes, and increased efficiency and productivity in the wake of high turnover and low recruitment. However, researchers have posited that a hybrid approach to adopting robot technologies in the hospitality service environment has proven to be simultaneously advantageous for both customers and employees, underscoring the special considerations under multiple stakeholders in the industry [63]. For example, service robots and other AI technology could be deployed for highly automated tasks (i.e., vacuuming, delivering items to rooms, accepting payment), thus freeing hospitality employees to serve customers in ways that are more personally and occupationally fulfilling. As service quality is a tenet of the hospitality industry and a focal point of differentiation and competitive advantage, a shift to reskill workers to manage service robots and related technologies for repetitive tasks, will allow the hybrid approach to direct what resources remain after the pandemic, to focus on guest service quality.

These industry-specific trends do not align well with the more generalized, and somewhat anecdotal, prediction [59] of the disappearance of medium skill jobs and the use of reskilling interventions to target advancement to different, high-skill jobs [1]. Within the hospitality industry, tailored solutions exist, such as the hybridized reskilling approach, that better represent the predicted stability of the need for person-delivered service alongside advancing technologies. Several recommendations have been made in observance of reskilling employees to integrate advanced technology into medium-skill service roles and include the potential for hiring hardware and software specialists, training employees to manage teams of robots, and the reorganization of departments with a focus on services and marketing integrations for the use of robots [64]. Anecdotal industry reports advocate for the use of robots as a supplement to human workers, rather than a replacement [65]. Targeting the use of robots and other AI technologies for low-level repetitive tasks enable hospitality employees to work safely and more efficiently while focusing on their primary stakeholder, the guest, to deliver on the service promise. In other words, an industry-informed solution involves reskilling to work in the same job alongside technology, ideally using technology to take over dangerous or repetitive tasks, as opposed to the broadly-based solution that proposes reskilling with a completely different job in mind.

### 5.2. Work-Family Programming

Within discussions of the changing nature of work, trends in both the “workplace” and the “workforce” highlight the growing challenges of work-life balance. This is influenced by multiple future of work trends, as increasing rates of remote work blur the physical boundaries between work and home life and more equitable rates of labor force participation bring work-family challenges for dual-earner families [1]. Poor work-life balance is indeed an occupational health and safety challenge, as it is associated with psychological and behavioral strain [66]. Work–family programming represents a systemic organizational solution to this future of work challenge, and typically includes paid leave benefits (i.e., parental leave), resources dedicated to the management of family demands (i.e., employer-sponsored childcare), education for supervisors regarding the best ways to support work-life balance for their subordinates, and flexible work arrangements such as self-scheduling [1].

The future of work challenge of work-family conflict is relevant to the hospitality industry, yet the drivers are distinct. Rather than increases in remote work, factors such as “irregular hours, emphasis on face time, frequent relocation,” [67] low job security, and job tasks that require a high degree of coordination between shifts [68] contribute to work-family issues among hospitality employees. In fact, those in the service industries report worse work-life balance than employees who belong to other industries, even after controlling for work hours [69].

Similar to the generalized solutions mentioned above, successful programming to alleviate work-family conflict in the hospitality industry includes the presence of paid leave benefits, visible work-life balance behaviors from senior management, and the presence of resources such as childcare [70]. However, it is worth noting that these programs have historically focused on the retention of highly educated hospitality employees [70]. It seems that trends regarding benefits coverage in the hospitality trend may not have dramatically changed recently, as 2021 data reports that the average hospitality employee receives benefits that value USD3.10 per every hour worked [42], while the national average is USD 12.18 per hour worked [71]. These generalized solutions to work-family issues, when implemented alone, could ignore the needs of a larger proportion of the hospitality industry, specifically front-line service workers or contract workers in positions that do not typically receive comprehensive benefits. Thus, these examples illustrate a gap where effective broadly developed future of work solutions may not be applicable to all industry contexts, and where the context of implementation must be carefully considered.

In contrast, hospitality organizations have successfully implemented programs that are not typically mentioned in traditional work-family programming for the general workforce, such as job sharing to promote work-life balance [70]. In addition, although realistic job previews (RJPs) are discussed as a general best practice across workplaces, hospitality researchers have noted the increased importance of RJPs specifically geared towards the work-life context (i.e., previewing hours and scheduling to help workers proactively anticipate potential work-life issues) in the hospitality industry. That is, while RJPs can be broadly considered a turnover-reducing measure, in the hospitality industry they are regarded as an intervention related to both turnover and work-family conflict [70]. In these cases, the uses of job sharing and RJPs to specifically target work-life issues acknowledge the realities of the hospitality industry. Hotels are open 24 h per day, including all holidays when the general public does not work, and consumer demand peaks when many other industries’ business hours wane [70]. Were more generalized solutions to work-family conflict be implemented in the hospitality industry in these instances, those initiatives would likely not account for these constraints.

## 6. Moving towards a Context-Informed Future of Work

The paper addresses timely and important topics, with illustrative examples presented from an industry that is a priority for safety and health research and practice. However, we must also acknowledge that the conclusions in this paper could be limited by the scope of the content analysis performed and that the two illustrative examples presented may not represent every opportunity for industry-informed solutions. The ideas discussed in this paper have implications for both research and practice in OHP and broader OSH disciplines. From a research perspective, to bridge the gap between the current industry agnostic OHP literature and a future body of literature that intentionally incorporates industry context to address the changing nature of work challenges and solutions, it is important to identify potential facilitators of this shift. We assert that further attention to the differences between multidisciplinary, interdisciplinary, and transdisciplinary collaboration structures will be an important facilitator. Multidisciplinary research draws from multiple disciplines, interdisciplinary work is associated with interaction across disciplines (i.e., working in parallel), and transdisciplinary research transcends traditional disciplinary boundaries with each discipline informing the approaches, methods, and theories of the other [9]. We assert that as OSH-informing disciplines evolve to address the future of work, it is important that each discipline strike a balance of breadth and depth in their understanding of challenges and solutions.

While tremendous progress has been made in bringing research out of disciplinary silos, we believe that a transdisciplinary collaboration structure, with a greater degree of disciplinary integration in which each discipline actively informs the other through collaborative research, would facilitate the systematic study of context. A transdisciplinary team might develop more comprehensive and impactful research, weaving together both generalized work-related phenomena and the way industry context shapes a worker’s experience of this phenomenon, with benefits for richer theory and research across OSH disciplines.

In addition to these existing facilitating trends, we encourage OHP researchers to promote other conditions that may pave a way for more industry-informed research. For example, journal editors and reviewers may develop criteria to help facilitate a more frequent consideration of context. Whereas prestigious journals evaluate submissions based on theoretical and empirical contributions, an author could also craft a compelling argument for how context impacts theories, empirical findings, and practice. Similarly, extramural funding agencies could emphasize criteria for reviewers that assess both the generalized and context-specific features of a proposal and encourage transdisciplinary teams and proposed deliverables that are likely to promote shared knowledge across disciplines rather than outlets in disciplinary silos. Facilitators of context could be developed in more localized settings as well. Universities tend to regard collaborations with community partners as mutually beneficial [72], and as such could develop criteria for tenure and promotion committees or performance evaluators that weight a strong publication with contributions related to industry context on the similar level as a publication with more generalized contributions. Incorporating these facilitators into publication, funding, and evaluation systems will work to create an incentive structure that encourages contributions in both broad and specific domains.

In terms of implications for practice, recent reviews of international OSH trends reveal that ensuring global consistency in industry-contextualized regulation of work hazards should be a major priority. Workplace safety regulations only cover about 10% of hazardous industries in developing countries worldwide, and variation in regulations by industry and geographic region present a challenge for the protection of workers [73]. These authors also advocated for more frequent “inter—collaborations of external industries” in practice to protect worker safety [73]. Thus, although this paper presents a primarily U.S.-centric view of industry for the purposes of this Special Issue, there clearly value in ensuring widespread attention to industry context in practice.

Adding to the implications for both research and practice, we also argue that a continued focus on research-to-practice will facilitate greater attention to industry. As the OHP literature develops a more comprehensive portfolio of interventions to address safety and health concerns in the workplace [74], there will likely also be an increase in the need to replicate a successful intervention in another context, such as another industry. Studies that document the intervention adaptation process describe the steps by which and intervention is systematically modified to increase fit in a new context and evaluate its effectiveness [75]. Summaries of the practice of intervention adaptation recognize that an intervention represents a new event occurring within the pre-existing context of a system, and the intervention may need to be modified on geographic or sociocultural principles to promote transferability [76] or the available resources and competing demands of an organization or industry [77,78]. Strong academic–partitioner partnerships could facilitate the more widespread use of this practice by allowing researchers to select evidence-based intervention and refer to practitioner guidance for modification in the local context; however, more research is needed on industry-related factors that could necessitate modification.

## 7. Conclusions

Across OSH-informing disciplines, norms vary to the extent of incorporating industry as meaningful context when studying or improving workplace safety, health, and well-being. As OSH is pivoting to address the complexities of the changing nature of the workplace, workforce, and work itself, now is an appropriate time to better understand the role of industry in OSH research and practice, especially in OSH disciplines that have not typically considered industry. As illustrated by our selected examples of future of work challenges and solutions within the hospitality industry, OSH necessitates a tailored approach as opposed to a “one-size-fits-every-industry” approach. By better understanding the way industry shapes work tasks and settings, shared values and assumptions, and individual and group identities of contemporary importance, we can ready researchers and practitioners for the future of occupational safety and health with expertise informed by both breadth and depth.

## Data Availability

Not applicable.

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
