# Peer review of "Industry Context as an Essential Tool for the Future of Healthy and Safe Work: Illustrative Examples for Occupational Health Psychology from the Hospitality Industry"

_ijerph, 2021, doi:10.3390/ijerph182010720_

Round 1
Reviewer 1 Report
The purpose of this manuscript is to present the current state of knowledge regarding the relationship between occupational health psychology and workplace health and safety issues. The issue is not new and the authors need to present their arguments regarding the new information generated by this work.
The introduction should be revised; it should begin with general issues and progress to the purpose of this work. For example, too soon, the reader is confronted with unsupported statements that must be supported by references and explained. Currently, the reader may be unconvinced, which is critical in a manuscript. For example, the introduction's backbone (lines 35-53) contains only two references. Furthermore, the title, abstract, and purpose of the paper are all partially deceptive. This is a paper about the hospitality industry, which should be more clearly presented and reflected in the manuscript. Aside from that, I am confident that the authors will be able to revise their work and discuss the broader issues of their goal in the final sections of their manuscript and highlight the originality of their work.
Author Response
Reviewer One:
Reviewer One, Comment One: The purpose of this manuscript is to present the current state of knowledge regarding the relationship between occupational health psychology and workplace health and safety issues. The issue is not new and the authors need to present their arguments regarding the new information generated by this work.
Response to Reviewer One, Comment One: We are very appreciative of your comments and recommendations. Regrettably, we did not communicate the purpose of this paper in a clear enough manner. The purpose of the paper was not to present the current state of knowledge regarding the relationship between occupational health psychology and workplace health and safety issues. Instead, it was to argue that industry context, and often neglected variable in occupational health psychology, is essential to address future of work challenges in occupational health psychology. To make the purpose of the paper clearer, we have added the following to the end of the first paragraph:
“In this paper, we explore applications of industry context to promote both breadth and depth in OSH disciplines as they address future of work challenges. We describe future of work challenges and solutions that require an appreciation of context within an illustrative group, the hospitality industry, and in doing so hope to motivate OHP researchers and practitioners to incorporate a focus on the nuance provided by industry more often. “
Reviewer One, Comment Two: The introduction should be revised; it should begin with general issues and progress to the purpose of this work. For example, too soon, the reader is confronted with unsupported statements that must be supported by references and explained. Currently, the reader may be unconvinced, which is critical in a manuscript. For example, the introduction's backbone (lines 35-53) contains only two references.
Response to Reviewer One, Comment Two: We are very appreciative of this suggestion; However, we feel that given that we did not communicate the purpose of the paper clearly, that the existing organization of the paper will be more clearly relevant now that the purpose of the paper is better stated. The organization progresses from general to specific with the following major sections:
- A section establishing the value of industry context and demonstrating that industry context is currently under addressed in OHP.
- A section proposing that industry context is necessary in order to addressed nuanced issues associated with future of work challenges.
- A group of sections describing the context of the hospitality industry and presenting examples in which generalized future of work solutions would fail without recognizing industry context in the hospitality industry
- A section that presents recommendations to promote industry-informed work in the future.
The point about the lack of citations in lines 35 – 53 is well taken. The World Health Organization model of Healthy Workplaces discusses the idea of the lack of a one size fits all solution in detail, and therefore we have added this important citation to this section.
Reviews One Comment Three: Furthermore, the title, abstract, and purpose of the paper are all partially deceptive. This is a paper about the hospitality industry, which should be more clearly presented and reflected in the manuscript.
Response to Reviewer One, Comment Three: In an effort to better represent the purpose of the paper, we have changed the tile to: “Industry Context as an Essential Tool for the Future of Healthy and Safe Work: Illustrative Examples for Occupational Health Psychology from the Hospitality Industry.”
Reviewer One, Comment Four: Aside from that, I am confident that the authors will be able to revise their work and discuss the broader issues of their goal in the final sections of their manuscript and highlight the originality of their work.
Response to Reviewer One, Comment Four: We appreciate your encouraging words!

Reviewer 2 Report
IJERPH – Industry Context as an Essential Tool for the Future of Healthy and Safe Work: Insight for Occupational Health Psychology from the Hospitality Industry
IJERPH 1384249
An interesting and informative paper, which attempts to account for the value of industry context and recommendations for achieving both breadth and depth of scientific inquiry and practical reach with a specific emphasis using examples from the hospitality industry to describe challenges or solutions to future of work issues that would be better understood or implemented.
From this perspective, the paper will garner interest from academia. From an academic perspective the paper presents a broad mix of current peer reviewed research and therefore, I would now consider the paper robust with ample depth, breath and scope of perspective in terms of currency, and in the discussion of divergent and conflicting perspectives. I see no shortcomings in the paper; and from a professional academic perspective, I believe my fellow colleagues in the academic community will consider the paper as having comprehensiveness and currency. One opportunity for growth within the paper is for the author(s) to use a consistent standard for citations – in some instances, there is a use of numbers following the cited author(s), in other places there is a use of author(s) names and date. Such an approach will create confusion for readers, and detracts from the quality of the paper; and will impact readability for some readers.
From a methodology perspective, the author(s) did not identify limitations; and an academic perspective, I would consider this a shortcoming since such an inclusion will enable the reader audience to understand the limitations and any caveats which impact or have an influence related to the research. As well, the inclusion of limitations will enable reader(s) to have the opportunity to place the paper into a context in relation to what is stated / proposed by the author(s).
Overall, the examples used in the paper are presented in a clear and concise manner; there is evidence of analysis. Additionally, the author(s) have developed a linkage between the literature, the examples used, and conclusions noted. Furthermore, the conclusions to the paper are tied together into a final coherent picture. It is evident that the author(s) have an excellent understanding of the subject area.
Overall, this is a solid paper – it provides several opportunities for continued research in the subject area with the possibility of different streams within the research area, while providing further avenues of research potential. With respect to the practical application of the research, it presents an opportunity to enhance the depth, breadth and understanding of how to achieve both breadth and depth of scientific inquiry and practical reach to account for the value within an industry context.
The writing quality of the paper is sound and logical, there is a solid scholarly and academic quality to the paper. – the author(s) are commended for taking steps to focus the writing of paper to ensure grammatical correctness.
Thank you for opportunity to review your interesting and informative article – it is well developed, well written and merits publication. I do, however, recommend several minor revisions. First, I recommend you use a consistent standard for citations – in some instances, there is a use of numbers following the cited author(s), in other places there is a use of author(s) names and date. Such an approach will create confusion for readers, and detracts from the quality of the paper; and will impact readability for some readers. Secondly, from a methodology perspective, you did not identify limitations; and an academic perspective, I would consider this a shortcoming since such an inclusion will enable the reader audience to understand the limitations and any caveats which impact or have an influence related to the research. As well, the inclusion of limitations will enable reader(s) to have the opportunity to place the paper into a context in relation to what is stated / proposed by the author(s).
I hope you will take the time to incorporate the suggestions revisions to take your paper to the next step – publication.
Author Response
Reviewer Two:
Reviewer Two, Comment One: An interesting and informative paper, which attempts to account for the value of industry context and recommendations for achieving both breadth and depth of scientific inquiry and practical reach with a specific emphasis using examples from the hospitality industry to describe challenges or solutions to future of work issues that would be better understood or implemented. From this perspective, the paper will garner interest from academia. From an academic perspective the paper presents a broad mix of current peer reviewed research and therefore, I would now consider the paper robust with ample depth, breath and scope of perspective in terms of currency, and in the discussion of divergent and conflicting perspectives. I see no shortcomings in the paper; and from a professional academic perspective, I believe my fellow colleagues in the academic community will consider the paper as having comprehensiveness and currency.
Response to Reviewer Two, Comment One: Thank you very much for your encouraging comments!
Reviewer Two, Comment Two: One opportunity for growth within the paper is for the author(s) to use a consistent standard for citations – in some instances, there is a use of numbers following the cited author(s), in other places there is a use of author(s) names and date. Such an approach will create confusion for readers, and detracts from the quality of the paper; and will impact readability for some readers.
Response to Reviewer Two, Comment Two: Thank you for this suggestion. The alternating use of citation formats was a regrettable error that has been corrected in this submission.
Reviewer Two, Comment Three: From a methodology perspective, the author(s) did not identify limitations; and an academic perspective, I would consider this a shortcoming since such an inclusion will enable the reader audience to understand the limitations and any caveats which impact or have an influence related to the research. As well, the inclusion of limitations will enable reader(s) to have the opportunity to place the paper into a context in relation to what is stated / proposed by the author(s).
Response to Reviewer Two, Comment Three: Thank you for this helpful suggestion. We have added text regarding strengths and limitations on line 373.
Reviewer Two, Comment Four: Overall, the examples used in the paper are presented in a clear and concise manner; there is evidence of analysis. Additionally, the author(s) have developed a linkage between the literature, the examples used, and conclusions noted. Furthermore, the conclusions to the paper are tied together into a final coherent picture. It is evident that the author(s) have an excellent understanding of the subject area.
Overall, this is a solid paper – it provides several opportunities for continued research in the subject area with the possibility of different streams within the research area, while providing further avenues of research potential. With respect to the practical application of the research, it presents an opportunity to enhance the depth, breadth and understanding of how to achieve both breadth and depth of scientific inquiry and practical reach to account for the value within an industry context. The writing quality of the paper is sound and logical, there is a solid scholarly and academic quality to the paper. – the author(s) are commended for taking steps to focus the writing of paper to ensure grammatical correctness.
Thank you for opportunity to review your interesting and informative article – it is well developed, well written and merits publication. I do, however, recommend several minor revisions. First, I recommend you use a consistent standard for citations – in some instances, there is a use of numbers following the cited author(s), in other places there is a use of author(s) names and date. Such an approach will create confusion for readers, and detracts from the quality of the paper; and will impact readability for some readers. Secondly, from a methodology perspective, you did not identify limitations; and an academic perspective, I would consider this a shortcoming since such an inclusion will enable the reader audience to understand the limitations and any caveats which impact or have an influence related to the research. As well, the inclusion of limitations will enable reader(s) to have the opportunity to place the paper into a context in relation to what is stated / proposed by the author(s).
I hope you will take the time to incorporate the suggestions revisions to take your paper to the next step – publication.
Response to Reviewer Two, Comment Four: Our team is so appreciative of your helpful review!

Round 2
Reviewer 1 Report
Thank you for your effort to accommodate all comments and suggestions your manuscript is now significantly improved and in my opinion contributes in this field
I suggest to delete the 1st sentence ("This paper is not ....") of section 6